# Evaluation of Cell Migration and Cytokines Expression Changes under the Radiofrequency Electromagnetic Field on Wound Healing In Vitro Model

**DOI:** 10.3390/ijms23042205

**Published:** 2022-02-17

**Authors:** Erica Costantini, Lisa Aielli, Federica Serra, Lorenzo De Dominicis, Katia Falasca, Pamela Di Giovanni, Marcella Reale

**Affiliations:** 1Department of Medicine and Science of Aging, University “G. d’Annunzio”, Via dei Vestini, 66100 Chieti, Italy; erica.costantini@unich.it (E.C.); katia.falasca@unich.it (K.F.); 2Department of Innovative Technologies in Medicine and Dentistry, University “G. d’Annunzio”, Via dei Vestini, 66100 Chieti, Italy; lisa.aielli@unich.it (L.A.); dedominicislorenzo2@gmail.com (L.D.D.); 3Department of Pharmacy, University “G. d’Annunzio”, Via dei Vestini, 66100 Chieti, Italy; federica.serra@unich.it (F.S.); pamela.digiovanni@unich.it (P.D.G.)

**Keywords:** RF-EMF, wound healing, keratinocytes, cytokines, MMPs

## Abstract

Wound healing (WH) proceeds through four distinct phases: hemostasis, inflammation, proliferation, and remodeling. Impaired WH may be the consequence of the alteration of one of these phases and represents a significant health and economic burden to millions of individuals. Thus, new therapeutic strategies are the topics of intense research worldwide. Although radiofrequency electromagnetic field (RF-EMF) has many medical applications in rehabilitation, pain associated with musculoskeletal disorders, and degenerative joint disorders, its impact on WH is not fully understood. The process of WH begins just after injury and continues during the inflammatory and proliferative phases. A thorough understanding of the mechanisms by which RF-EMF can improve WH is required before it can be used as a non-invasive, inexpensive, and easily self-applicable therapeutic strategy. Thus, the aim of this study is to explore the therapeutic potential of different exposure setups of RF-EMF to drive faster healing, evaluating the keratinocytes migration, cytokines, and matrix metalloproteinases (MMPs) expression. The results showed that RF-EMF treatment promotes keratinocytes’ migration and regulates the expression of genes involved in healing, such as MMPs, tissue inhibitors of metalloproteinases, and pro/anti-inflammatory cytokines, to improve WH.

## 1. Introduction

Most cutaneous lesions resolve through a progression of different phases including hemostatic, inflammatory, proliferative, and remodeling phases, taking one to two weeks to heal. However, this time is prolonged in chronic wounds, which do not progress systematically through the healing phases [1]. In impaired healing, wounds remain in a chronic inflammatory phase due to weakened cellular migration, growth factor release, and poor microcirculation, as observed in numerous conditions such as acute and chronic diseases, aging, obesity, smoking, nutritional deficiencies, or after surgery [2]. In addition, in chronic wounds, various host microbes colonize and multiply within the unhealed tissue, further contributing to impaired healing [3]. Consequently, impaired healing is susceptible to complications that also have a negative impact on patients themselves and on health care costs, in addition to contributing towards decreased productivity and a reduction of work time [4]. For the proper progression of wound healing (WH) phases, once the critical point is reached, the current phase must have the ability to turn itself off and promote the next phase. The mechanism by which different phases inhibit and promote each other is through the expression of soluble mediators, such as cytokines and degradative matrix metalloproteinases (MMPs) that, by guiding interconnected cycles of feedback/forward, allow for an orderly progression of WH. Uncontrolled inflammation prevents, rather than promotes, WH and, in impaired healing, the continued up-regulation of inflammation leads to an abnormal inflammatory profile characterized by an imbalance between MMPs and their inhibitors, favoring wound degradation [5,6]. During wound repair, inflammation and epithelialization are mutually dependent and related processes. The prevalence of inflammatory cytokines with respect to pro-proliferative factors causes a short-circuit of inflammation. WH remains in the inflammatory phase instead of progressing towards the proliferative phase, when persistent inflammation drives further cell damage, causing more inflammation and hampering the advancement toward a healed wound. Thus, the extent of wound inflammation appears to inversely correlate with the re-epithelialization rate [7].

The research to develop effective therapeutic treatments for promoting fast WH after an injury is crucial, and this study highlighted the new therapeutic perspective to improve cutaneous healing and tissue regeneration.

The use of electromagnetic fields (EMF) in therapy is rapidly spreading. In our previous studies, we demonstrated that extremely low-frequency (ELF)-EMF (50 Hz, 1 mT) accelerates WH modulating inflammatory mediators and keratinocyte proliferation/migration [8]. Several devices have been developed to generate non-ionizing and non-thermal low-frequency (LF) pulsed electromagnetic fields (PEMF) with specific waveforms, amplitudes ranging between 6 and 500 Hz, and magnetic flux densities between 0.1 mT and 30 mT, that induce bioelectric currents in tissues, producing special biological effects [9]. A recent meta-analysis assessing the effectiveness of radiofrequency (RF)-PEMF therapy in clinical studies found statistical support for the efficacy of the therapy in treating post-operative pain and edema [10].

Bearing in mind that specific time-varying exposure to an electromagnetic field can affect biological systems differently [10], we hypothesized that the ability of RF to interact with biological processes is dependent on the temporal patterns of the fields and that a short-term exposure interacts with cells and activates molecular pathways leading to cell activation, differentiation, and the release of mediators, promoting a shift towards subsequent WH phases. However, protracted exposure may be responsible for the reduction of inflammation and pain [11]. Thus, we designed three different experimental protocols to identify the specific pattern of exposure necessary to improve WH. Although repair involves several different cell types in sequential steps, keratinocytes are implicated in the intricate mechanisms of initiation, maintenance, and fulfillment of WH. Migration of epidermal keratinocytes, over the wound site, produce multiple factors to promote re-epithelialization, stimulate angiogenesis, and the generation of a connective tissue matrix [12].

Notoriously the levels of MMPs, that are low in normal skin, strongly increase after injury to modulate keratinocytes motility, allowing for re-epithelialization. In addition, MMPs regulate the inflammatory response by induction or inhibition of cytokines and growth factors, playing a key role during all phases of WH. High levels of MMP-2 and MMP-9 and low levels of tissue inhibitors of metalloproteinase (TIMP)1 are present in chronic wounds, and their inhibition can help to improve the healing process. MMP-9 creates a state of persistent inflammation and tissue destruction by digestion of fibronectin in fragments, but if the levels of MMP-9 are lower than normal, epithelialization may be wavered [13]. Moreover, MMP-13 expressed in the chronic wound bed is involved in the WH process from the early stages of granulation tissues production to the induction of inflammation, angiogenesis, and matrix degradation [14].

We used the human keratinocyte cell line (HaCaT) to investigate the influence of 27.1 MHz (37 ns) RF-EMF, which are commonly applied by commercial devices, on mediators which drive the progression from the inflammatory to the proliferative phase and subsequently to the remodeling phase of wound repair, analyzing the effects of three different experimental protocol setups.

Our data describe, at different setup exposures, that RF-EMF had different effects on the expression of inflammatory mediators and on the migration of HaCaT cells, suggesting new benefits in the development of novel therapeutic approaches to accelerate healing processes.

## 2. Results

### 2.1. RF-EMF Exposure Does Not Change the Viability of HaCaT Cells

The effect of RF-EMF on cell growth was determined by the measurement of metabolic activity after 24 h of exposure. As demonstrated in Figure 1, in RF-EMF-exposed cells, for different exposure conditions, cell vitality was not significantly affected.

### 2.2. RF-EMF Modulates Oxidative Balance in HaCaT Cells

A fine balance between the positive and deleterious effects of ROS is required during correct WH. Owing to the high levels of ROS present in wounded cells, the expression of ROS-detoxifying enzymes in healing wounds appears to be of particular importance. Thus, we evaluated both H_2_O_2_ levels and ROS-detoxifying GSH enzyme, the main metabolites involved in determining the cellular redox state. Moreover, GSH levels are important for assessing toxicological responses during RF-EMF exposure.

ROS were measured through the H_2_O_2_ concentration within the cell cultures. As reported in Figure 2a, levels of H_2_O_2_ were increased in RF-EMF-exposed cells, although only in Protocol C exposure conditions, which exhibited a significant difference with respect to the sham exposure. GSH levels are higher after each protocol setup, to maintain the equilibrium between free radicals and antioxidants (Figure 2).

### 2.3. RF-EMF Promotes Proliferation and Migration of HaCaT Cells in a Model of Wound Healing

During WH, keratinocytes proliferate and migrate closing the wound and restoring the epithelial layer [15]. To test if there were quantifiable migration rate differences between keratinocytes exposed to different timings of RF-EMF and the sham-exposed keratinocytes in response to wounding, a cell migration assay was performed. HaCaT were grown until confluence and scratch wounds were then created using a pipet tip and the wound site was photographed over time (T0, T6, and T24). The cell-free area was evaluated, and the cell migration rate was calculated as a reduction of cell-free area. A reproducibility test showed that repeated annotation of the same image produced an average difference of the total image area between replicates that never exceeded 5%. Our data demonstrated that RF-EMF contributed to progressive wound closure in a time-dependent manner. Quantification of the wounded area revealed that HaCaT cells, exposed to RF-EMF, during the initial 6 h of incubation (Protocol B) migrated faster to the wounded area than in the control, and also in Protocol A and C-exposed cells, compared to the sham-exposed cells. By 24 h after the scratch, the cell-free area was completely covered in Protocol B-exposed cells (Figure 3a,b). Moreover, to investigate if RF-EMF’s ability to promote wound closure is due to cell migration or proliferation, we evaluated cell proliferation in the presence of mitomycin C, the cell proliferation inhibitor. Our results showed that the wound closes though cell viability is reduced. Thus, exposure to RF-EMF induces cell migration and gap closure (Figure 3c).

### 2.4. RF-EMF Modulates Expression of Inflammatory Mediators in HaCaT Cells

To evaluate how the different RF-EMF exposure setups modulate inflammatory mediators, we analyzed their expression after each exposure condition in our in vitro model. After 30 min exposure, 6 h without exposure, and then another 30 min of exposure (Protocol A) we observed a significant increase of tumor necrosis factor (TNF)α and transforming growth factor (TGF)β gene expression, and their higher expression was also detected after 6 h of exposure (Protocol B), relative to the sham-exposed cells. While in the HaCaT exposed continuously for 24 h (Protocol C), TGFβ and TNFα were lower than in the HaCaT sham-exposed for 24 h. Interleukin (IL)-18 and Cyclooxygenase (COX)-2 expressions were reduced in Protocol A and Protocol B, and unchanged in Protocol C exposure, relative to the control. IL-4 expression was greater after all protocol exposure conditions in accord with its role in the induction of the TGFβ-producing M2 phenotype. Moreover, TNFα and TGFβ were more induced by exposure Protocol B, accelerating the transition from the inflammatory to healing phase (Figure 4a–c).

### 2.5. RF-EMF Modulates Expression and Production of IL-6

As an important modulator of the inflammatory and reparative process, IL-6 plays a key role in keratinocytes’ differentiation, activation, and proliferation. As shown in Figure 5, we evaluated its expression and production after different exposure conditions. After 6 h of exposure (Protocol B), both expression and production were significantly increased compared to sham-exposed cells, in accordance with IL-6’s ability to switch from an inflammatory to a reparative phase. Furthermore, lower levels of IL-6 were observed in Protocol C-exposed cells (Figure 5).

### 2.6. RF-EMF Modulates Expression of MMPs in HaCaT Cells

The MMP-2 and MMP-9 expression is linked with increased keratinocyte migration, and the balance between MMPs and their inhibitors plays a pivotal role in regulating extracellular matrix degradation and deposition that are essential for successful WH. Thus, we evaluated MMPs expression levels, 24 h after scratch, in HaCaT cells exposed at different exposure setups. As reported in Figure 6a, in exposure Protocol A, MMP-9 and MMP-2 were higher, although not significantly, than in the sham-exposed cells, maybe for the need to depredate the matrix. MMP-13, with proteolytic activity, and TIMP1, a metal-peptidase inhibitor, are reduced. After 24 h of wound scratch and Protocol B exposure, we observed a reduction of MMP-9, a not significant change in MMP-2, and a decrease for MMP-13 and TIMP1, although the TIMP1 result was increased compared to the exposure condition A (Figure 6b). In Protocol C-exposed cells, the gene expression is comparable with that observed in condition B, except for the increase in MMP-9, probably linked to the incomplete wound closure (Figure 6c).

## 3. Discussion and Conclusions

In the microenvironment of the damaged tissue, numerous biological factors and cell types are involved. One of them is the keratinocytes, which play a key role in early response to wounding and release of proinflammatory cytokines that are important for initiating the healing response and then also during the re-epithelialization phase. Acting in an autocrine and paracrine way, cytokines induce keratinocytes’ proliferation and migration to the wounded area [16,17], determining a positive-feedback for proinflammatory cytokines release, including IL-6 and TNFα. The inflammatory phase in WH is considered to be a preparatory process for the formation of new tissue. After an injury, the rapid activation of TNFα drives the stimulation of keratinocyte and fibroblast proliferation, modulation of the immune response, and synthesis and breakdown of extracellular matrix proteins. Later, TNFα also promotes epithelial regeneration by canonical nuclear factor-kB (NF-kB)-dependent IL-6 activation [18]. The timing of the inflammatory response is paramount to a successful resolution and wound closure, and the role of IL-6 in the WH process is crucial. Early release of IL-6 promotes a pro-inflammatory response, whereas late release induces a reparative, anti-inflammatory response [19]. IL-6 acts through the signal transducer and activator of transcription (STAT)3-dependent pathway, which allows keratinocytes to respond to mitogenic factors and stimulate cell migration, and the impairment of the IL-6 signaling pathway delays WH [20]. IL-6 stimulates Th2 cells to produce IL-4 which promotes the transition from the proinflammatory M1 phenotype to the reparative and TGFβ-producing M2 phenotype of macrophages. TGFβ1 can promote keratinocytes’ migration by stimulating MMPs that facilitate cell migration by enabling cell-matrix detachment, an essential aspect of wound repair [21]. The balanced presence of MMPs and TIMP1 is necessary for keratinocytes’ migration; indeed, in impaired healing, the reduced migration is linked to high production of MMP-9 and reduced expression of TIMP1 [22].

To promote the WH, several EMFs with different frequencies, square waveform, and intensity have been explored [8,23,24,25,26,27,28]. In fact, several studies have evaluated the effects of PEMF on chronic diabetic foot ulcers but using different PEMF parameters and with contradictory findings [29,30], suggesting that combinations of intensity and treatment period may produce different effects on extracellular matrix synthesis and remodeling, cell proliferation and migration. However, further research is needed to identify the best timing and intensity before PEMF clinical applications.

The RF-EMF is non-ionizing energy at the shortwave radiofrequency band of the electromagnetic spectrum, commonly at a frequency lower than 30 MHz, that may be used as a non-invasive therapy delivered through a wound’s dressing using small and portable devices that can be self-applied. Recently, using up-to-date technology, the effectiveness of small wearable EMF devices in the reduction in post-operative pain has been demonstrated [31], through reducing inflammation with the consequent functional recovery of the affected area.

Proinflammatory signaling is a critical regulator of the WH process and, if altered, can result in wounds that take much longer to heal and that are at risk of infection, causing distress and requiring careful management. The goal of this study was to elucidate the efficacy and the mechanisms of RF-EMF-induced effects during WH. We investigated modulating effect of RF-EMF using three different exposure protocols and analyzing the timing exposure-dependent inflammatory mediators, MMPs, and redox homeostasis, via a widely used in vitro, wound scratch assay. This model is useful to study inflammatory and epithelialization phases of WH and allows for the evaluation of cell migration from the wound edge over the scratched area.

Our results showed that in RF-EMF treatment for 6 h (Protocol B) healing was significantly speeded up compared to the sham-exposed cells, in which WH was incomplete. These data provide proof of principle that RF-EMF interventions are effective at promoting WH, in part, by modulation of cytokines balance. TGFβ has a paradoxical effect in inflammation with both pro and anti-inflammatory activity; indeed, low levels of TGFβ are reported to be involved in chronic wounds, while they are increased during acute WH [32,33]. Consistent with Jeong et al. [34], we detected significantly higher levels of TGFβ after RF-EMF exposure Protocol B, with higher keratinocytes migration and faster healing of wounds, showing that TGFβ may promote keratinocytes’ migration through a mechanism involving integrin induction and small mother against decapentaplegic (Smad)-dependent signaling.

In addition to its inflammatory function, COX-2 is essential for normal WH and is stimulated in the early phase of repair. In our experiments, in accordance with the previous study, COX-2 seems to be linked to the production of the TGFβ, through the autocrine/paracrine effect [35].

In relation to its role as mediator of the inflammatory response and trigger factor for the adaptive immune activation [36], IL-6 is up-regulated in RF-EMF-exposed cells and, observing the wound closure rate after 24 h, we can suggest that an early IL-6 expression and production is a necessary factor in fast WH.

Keratinocytes, by the production of cytokines and growth factors, can control the activity of MMPs at transcriptional and post-transcriptional levels, and their expression increase early after cutaneous injury. In this study, we focused on MMP-2 and MMP-9, produced by migrating keratinocytes, that are associated with inflammation and remodeling. In healing impairments, the unbalanced production of MMPs and TIMP1 were involved. The results demonstrated that, after 24 h from the scratch, both MMP-2 and MMP-9 are higher in keratinocytes exposed to Protocol A and lower in keratinocytes exposed to Protocol B, according to the healing rate. Exposure to RF-EMF may help the healing of “hard to heal” wounds, modulating the network between cytokines, growth factors, MMPs, and TIMPs.

Low levels of ROS are essential in stimulating effective WH [37] whereas, besides their beneficial role in microbial killing, elevated ROS have been associated with cellular damage and impaired wound repair in chronic, non-healing wounds [37,38,39]. In fact, ROS can lead to sustained proinflammatory cytokines’ secretion, induction of MMPs, and impaired dermal fibroblast and keratinocyte function [40]. Chronic wounds are often characterized by the presence of excessive ROS or the absence of antioxidant ROS scavenger molecules, such as GSH. It has been suggested that reduced or delayed WH may be a consequence of an altered equilibrium between free radicals and antioxidants allowing the wound to proceed unchecked. In fact, in the elderly subject, WH was delayed, and interestingly, the levels of wound antioxidants decrease with aging [23].

Our results showed that in keratinocytes, exposure to RF-EMF improves their capacity to detoxify superoxide radical anions and that the increasing trend of ROS levels was counter-balanced by the increase of GSH levels in the in vitro wound-repair process, indicating that no damage results from RF-EMF exposure, so the cells have stabilized and normalized cell function which may ultimately accelerate WH. It is known that EMF setup, cell type, and growth cycle may determine different effects [41,42] and we support the hypothesis of using RF-EMF to promote the healing of wounds.

Results from this study clearly demonstrate that the duration of exposure plays a very important role in determining the effect of RF on cells in vitro, in fact, changes in signal duration and sequencing ensure a variety of biological processes such as RNA synthesis, protein expression and cell migration needed for tissue repair or regeneration. In this study we highlight that the dynamics of cellular responses, such as cell migration and cytokine expression, are concurrent with RF-EMF enhanced WH. Numerous cellular studies have focused on the effects of EMF on signal transduction pathways, with the calcium (Ca^2+^)-dependent signaling pathways receiving the most attention [43,44]. Our results could be consistent with the EMF effect on Ca^2+^-dependent signaling, which is known to up- and down-regulate inflammatory mediators in challenged biological systems. We elucidated the efficacy and the molecular mechanisms of RF-EMF induced effects and, in line with other studies, we showed that RF-EMF may favor an adaptative response to mechanical injury, leading to gene expression changes of the factors involved in the regulation of the inflammatory phase and driving the shift towards the re-epithelialization phase. The ease of use and compatibility with current conventional therapy, suggests that wearable RF-EMF devices could be widely applied as a first-choice therapy. Since changes in the duration, sequentiality, and temporal convergence of incoming signals ensure a multiplicity of biological outcomes, our in vitro results may assist with the development of a better understanding of the different biostimulatory or bioinhibitory reactions in respect of the different timing of RF applications, and they may also be useful in the choice of timing for in vivo therapeutic applications, although further controlled studies are required to determine their true value.

## 4. Materials and Methods

### 4.1. Cell Culture

HaCaT cells were obtained from the Cell Lines Service (CLS, cat. no. 300493). Cells were grown in Dulbecco’s modified Eagle’s medium (DMEM) supplemented with 10% fetal bovine serum and 1000 U penicillin, streptomycin (Sigma, Milano, Italy). Cell cultures were maintained at 37 ± 0.3 °C in an incubator with a humidified atmosphere of 5% CO_2_ (HeraCell 150, ThermoFisher Scientific, Waltham, MA, USA).

### 4.2. RF-EMF Exposure System

The exposure system for in vitro RF radiation is composed of a circuit powered by direct current, delivering a square wave emitted to packages of sinusoids, in pulsed RF, with a power <3 W. The device emits non-ionizing RF radiation at a carrier frequency of 27.1 MHz (37 ns). The carrier RF comes modulated through a pulse at 600 Hz (1.66 ms) duty cycle 10%. The single pulse is 167 µm. The exposure signal was fed through a copper antenna powered by a CR2032 lithium battery with a nominal voltage of 3 V, ensuring continuous functionality. The output of the RF-EMF is like the parallel between a 5 Ω resistor and a condenser from 150 pF. The external dimensions of the RF-EMF antenna were 6 mm × 12 mm.

### 4.3. Experimental Protocols

HaCaT cells were exposed to an RF-EMF device, fully characterized for electromagnetic and thermal dosimetry, and validated prior to the experiment. During the RF-EMF exposure, the temperature of media in the culture chamber was checked using a two-channel thermometer (TM-925, Lutron, Coopersburg, PA, USA). No significant temperature changes were observed to be associated with the application of the device. All experiments were performed in triplicate. In accordance with the timing of the in vivo application of RF-EMF devices, our objective was to evaluate intermittent, acute, or protracted exposure conditions, each with its own sham control (Figure 7).

In Protocol A, the cells were exposed for 30 min to RF-EMF, followed by 6 h to sham exposure and 30 min to a second RF-EMF exposure. The control was the sham-exposed conditions of keeping the cells for the same time but in the absence of any radiation.

In Protocol B, the cells were exposed for 6 h to RF-EMF. The control was the sham-exposed condition, keeping the cells in the same conditions as that of the exposed samples but in the absence of any radiation.

In Protocol C, the cells were exposed for 24 h to RF-EMF. The sham-exposed cells were kept under identical environmental and incubation time conditions as the RF-EMF-exposed cells, except for field exposure.

After each protocol setup and relative control, cell proliferation, migration, gene expression, and ROS/GSH were evaluated. In addition, biological parallels were used for each protocol type of treatment to test, also after 24 h of incubation, the cell proliferation, migration, gene expression, and ROS/GSH.

### 4.4. Cell Proliferation Assay

Cell proliferation rate was determined by the 3-(4,5-dimethylthiazol-2-yl)-2,5-diphenyltetrazolium bromide (MTT) assay, according to the recommendations of the manufacturer (Merck KGaA, Darmstadt, Germany). At the end of each incubation, MTT reagent was added for 3 h and incubated at 37 °C, and the absorbance was measured at OD 570 nm by the GloMax^®^-Multi Detection System (Promega Corporation, Madison, WI, USA).

### 4.5. Oxidative Stress

#### 4.5.1. GSH Assay

First, 1 × 10^5^ HaCaT cells were plated on white, clear-bottom 96-well plates with 100 μL of complete culture medium. After reaching confluence, cells were scratched and exposed, or not, to the RF-EMF device, for the different experimental times. To detect the oxidative stress induction, the intracellular production of GSH was evaluated using the GSH-Glo™ Glutathione Assay (Promega Corporation, USA), according to the manufactured protocol. Briefly, after incubation, the culture medium was replaced with GSH-Glo Reagent for 30 min, and luciferin detection reagent was subsequently added. Luminescence was measured after 15 min of incubation at room temperature. The glutathione detection and quantification were performed with reference to the standard GSH curve, prepared according to the indications of the kit. Luminescence readings of the plates were performed with the GloMax^®^-Multi Detection System (Promega Corporation, Madison, WI, USA).

#### 4.5.2. ROS Assay

The intracellular production of ROS was evaluated using the ROS-GloTM H_2_O_2_ kit (Promega Corporation, Madison, WI, USA). The test is based on the ability of the substrate to react directly with H_2_O_2_ to generate a precursor of luciferin. After the addition of ROS-Glo™ Detection Reagent containing Ultra-Glo™ recombinant luciferase and d-cysteine, the precursor was converted to luciferin by d-cysteine and the produced luciferin reacted with the Ultra-Glo™ recombinant luciferase to generate a luminescent signal which is proportional to the concentration of H_2_O_2_. Luminescence reading of the plates was performed with the GloMax^®^-Multi Detection System (Promega Corporation, Madison, WI, USA).

### 4.6. Mechanical Stretch Injury and Migration Rate Assays

HaCaT cells were seeded in a 6-well plate at 1.5 × 10^6^ cells/well in 1.0 mL of complete medium. When the cells reached 80%, the monolayer was gently scratched across the center of the well with a sterile pipette tip (Ø = 0.1 mm). The scratch determines the onset of secondary and progressive damage to the cells. Thus, we did not change the culture medium after wounding, keeping cell debris and other factors released from the detached cells. The scratch injured cells were incubated at 37 °C and 5% CO_2_ until wound closure. The WH was evaluated using phase-contrast images acquired using an inverted microscope (Leica DMi1, Wetzlar, Germany). Cell migration rate was calculated as the distance traveled by the cells, over time: just after the scratch (T0), after 6 h (T6), and after 24 h (T24). We calculated the change in a cell-free area, measuring the leading edges at the same four reference points for each well, using Leica LAS EZ Application Suite image analysis software. Three independent experiments were conducted, and four points of the wounded area were analyzed for each replicate. The percentage of the cell-free area was calculated referring to images at T0, for each sample. WH was compared between the control samples and the experimental samples by measuring the wound width.

### 4.7. Cell Proliferation Assay in Presence of Mitomycin C and Mechanical Scratch

HaCaT were plated at a density of 1 × 10^5^ in a 96-well plate, to evaluate the proliferation of the cells in the presence of mitomycin C and mechanical damage, under the different exposure conditions. When 90% confluence was reached, the culture medium was removed and replaced with a medium containing 10 μg/mL of mitomycin C, to inhibit cell proliferation. A mechanical scratch was performed in each well using a sterile tip (Ø = 0.1 mm). The cells were exposed for Protocol A-B-C or they were sham-exposed, as reported in the experimental design. The effect of the RF-EMF device on HaCaT cells proliferation was determined using an MTT assay (Merck KGaA, Darmstadt, Germany).

### 4.8. Real-Time PCR

Total RNA was isolated from HaCaT, using QIAzol reagent (Qiagen, Hilden, Germany), and reverse transcribed with QuantiTec Reverse Transcription kit (Qiagen, Hilden, Germany) according to the manufacturer’s instructions. Real-time qPCR was performed using GoTaq^®^ qPCR Master Mix (Promega, Milan, Italy) and a Bio-Rad Real-Time PCR instrument (CFX Real-Time PCR Bio-Rad, Hercules, CA, USA) with the following cycling conditions: 95 °C for 10 min, followed by 40 cycles of denaturation at 95 °C for 10 s, annealing at 60 °C for 10 s, and extension at 72 °C for 20 s. The primer sequences used for qPCR are provided in Table 1. RT-qPCR results were analyzed using Bio-Rad system software (CFX Manager). The 2^−ΔΔCt^ method was used to detect the relative expression of TNFα, TGFβ, IL-8, IL-18, COX-2, IL-4, IL-6, MMP-9, MMP-2, MMP-13, and TIMP1, using RPS18 to normalize the gene expression levels. Relative quantification cycle (Ct) values were reported as an expression fold changes. Experiments were performed in triplicate and the data were averaged [45].

### 4.9. IL-6 Enzyme-Linked Immunosorbent Assay

The concentration of IL-6 in the supernatant was obtained from the HaCaT cell culture, after treatment with the different experimental protocols, as determined by the Enzyme-Linked ImmunoSorbent Assay (ELISA). Specifically, after each experimental protocol, cell culture supernatant was collected and stored at −80 °C for subsequent evaluation. Before assessing the IL-6 levels, samples were centrifugated at 10,000× *g* for 5 min to eliminate cell debris and they were then plated following the manufacturer’s instructions. A Human IL-6 ELISA kit (Millipore, Merck KGaA, Darmstadt, Germany) was used, and relative absorbance was measured at 450 nm using the GloMax^®^-Multi Detection System (Promega Corporation, Madison, WI, USA). Cytokine concentration was calculated using a standard reference curve. The intra- and inter-assay reproducibility was >90%. The specificity and the sensitivity for the cytokine were defined according to the manufacturer’s guidelines.

## 5. Statistics

GraphPad Prism (v.6.0; GraphPad Software, La Jolla, CA, USA) was used for statistical analysis of the data. All results were expressed as mean ± standard deviation. For repeated measures, ANOVA was performed to compare the means between groups. The “fold change” of gene expression levels was calculated with the 2^−ΔΔCt^ method. The hypothesis that the fold change between exposure conditions (Protocol A-B-C) and the control, was equal to 1, was tested with Student’s t-test for unpaired data.

## Figures and Tables

**Figure 1 ijms-23-02205-f001:**
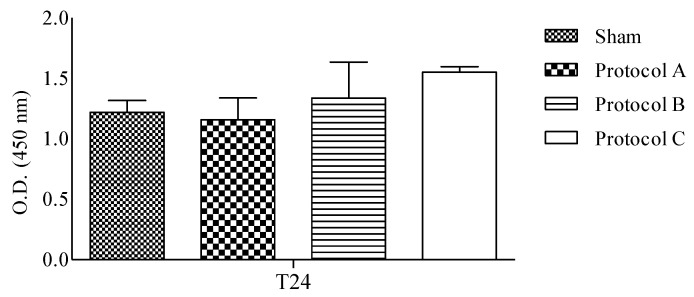
MTT assay. HaCaT proliferation rate was evaluated after each RF-EMF-exposure setup and the sham exposure. Combined data were from three experiments and are reported as mean ± SD of O.D. values.

**Figure 2 ijms-23-02205-f002:**
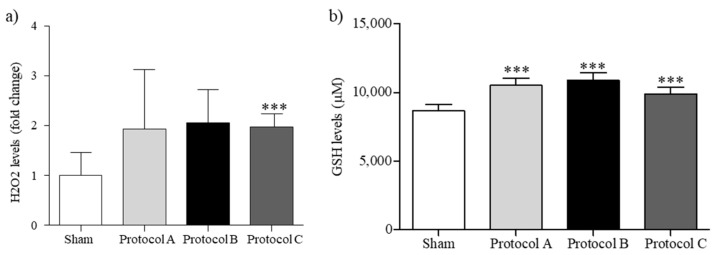
Oxidative Balance. (**a**) The H_2_O_2_ and (**b**) GSH levels change in HaCaT cells when exposed to RF-EMF. The results are expressed as a mean ± SD (*n* = 3). *** *p* < 0.001 vs. sham-exposed cells.

**Figure 3 ijms-23-02205-f003:**
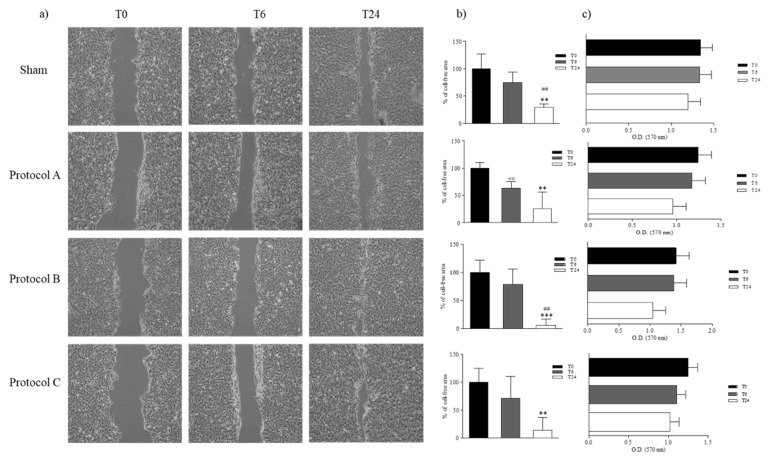
Wound healing assay. (**a**) Representative light microscope images of WH assays for HaCaT to evaluate migration rate at 6 and 24 h, after sham, Protocol A-B-C-exposure. Original magnification 10×. Scale bar = 0.5 cm. (**b**) Bar-graphs of the cell-free area over time, assuming 100% of the wound width just performed. Wound closure was evaluated by measuring the remaining cell-free area and expressed as a percentage of the initial cell-free area. The results of three independent experiments are expressed as mean ± SD of the percentage of the cell-free area. ** *p*-value < 0.01 in T24 vs. T0; *** *p*-value < 0.001 in T24 vs. T0; ## *p*-value < 0.01 in T24 vs. T6; °° *p*-value <0.01 in T6 vs T0. (**c**) MTT assay in the presence of mitomycin C and WH. Data are reported as mean ± SD of O.D. values obtained from triplicate experiments.

**Figure 4 ijms-23-02205-f004:**
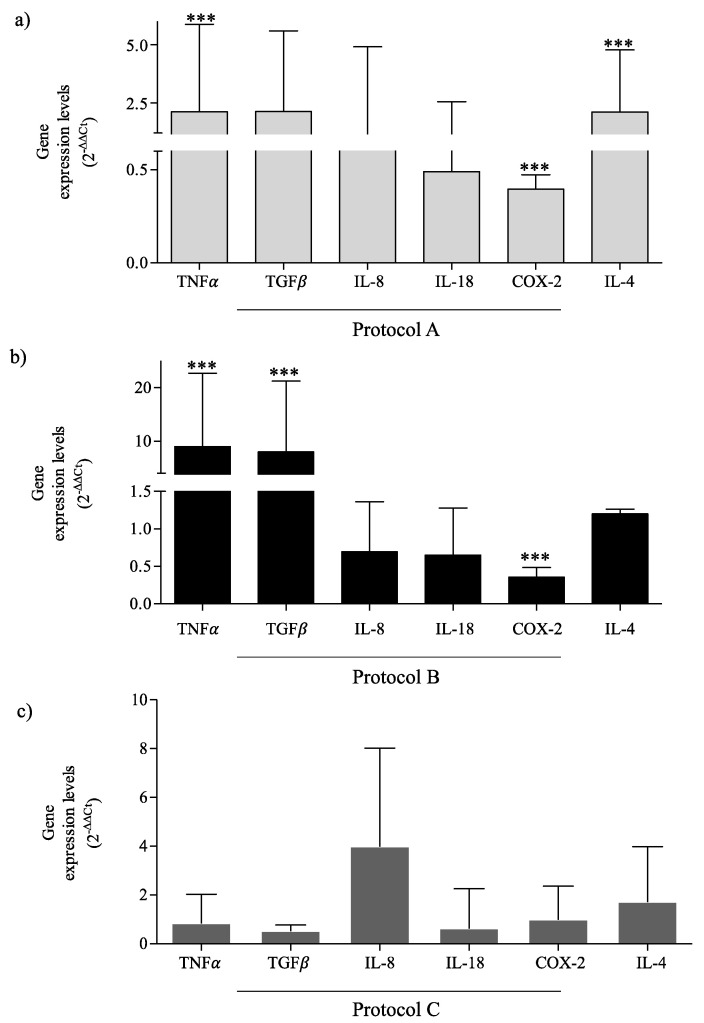
TNFα, TGFβ, IL-8, IL-18, COX-2, IL-4 gene expression. (**a**) Inflammatory mediators’ levels in Protocol A-exposed cells relative to sham-exposed cells. (**b**) Inflammatory mediators’ levels in Protocol B-exposed cells relative to sham-exposed cells. (**c**) Inflammatory mediators’ levels in Protocol C-exposed cells relative to sham-exposed cells. Data are reported as mean of 2^−ΔΔCt^ and 95% CI of three independent experiments. *** *p*-value < 0.001 vs. sham-exposed cells for the same time.

**Figure 5 ijms-23-02205-f005:**
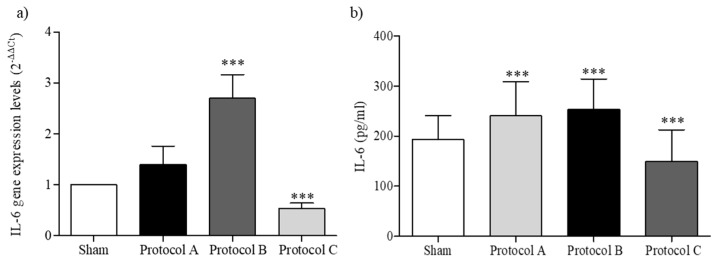
IL-6 expression and production. (**a**) IL-6 gene expression in HaCaT cells after different RF-EMF-exposure conditions. The mean values of gene expression levels (2^−ΔΔCt^) and 95% CI of three independent experiments, were reported. *** *p* < 0.001 with respect to sham-exposed cells. (**b**) IL-6 levels in HaCaT cell culture supernatant. The mean values ± SD were reported in triplicate experiments. *** *p* < 0.001 vs. sham-exposure condition.

**Figure 6 ijms-23-02205-f006:**
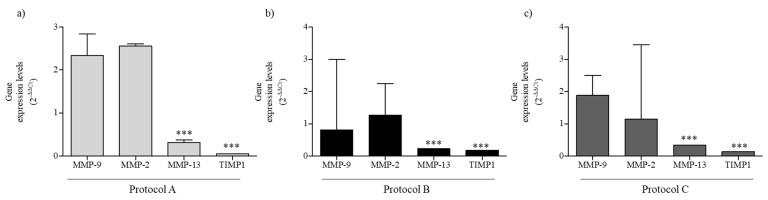
MMPs gene expression. MMP-9, MMP-2, MMP-13, and TIMP1 gene expression in HaCaT cells after different RF-EMF-exposure conditions. (**a**) Protocol A exposure; (**b**) Protocol B exposure; and (**c**) Protocol C exposure. The mean values of gene expression levels (2^−ΔΔCt^) and 95% CI of three independent experiments, were reported. *** *p* < 0.001 with respect to sham-exposed cells.

**Figure 7 ijms-23-02205-f007:**
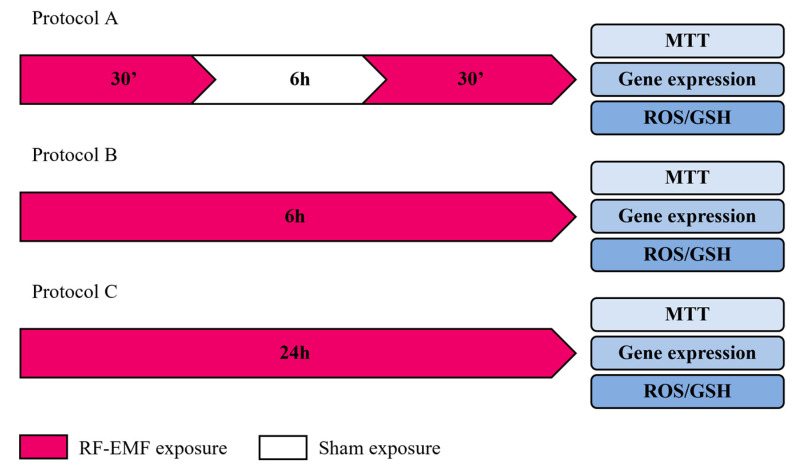
Experimental protocols.

**Table 1 ijms-23-02205-t001:** Gene sequences.

Gene	Forward Primer Sequence (5’-3’)	Reverse Primer Sequence (5’-3’)	Amplicon Lenght
** *TNFα* **	CCTTCCTGATCGTGGCAG	GCTTGAGGGTTTGCTACAAC	184 bp
** *TGFβ* **	AACAATTCCTGGCGATACCTC	GTAGTGAACCCGTTGATGTCC	197 bp
** *IL-8* **	GTGTAAACATGACTTCCAAGCTG	GTCCACTCTCAATCACTCTCAG	182 bp
** *IL-18* **	CAGTCAGCAAGGAATTGTCTC	GAGGAAGCGATCTGGAAGG	139 bp
** *COX-2* **	GACAGTCCACCAACTTACAATG	GGCAATCATCAGGCACAGG	105 bp
** *IL-4* **	CAAGTGACTGACAATCTGGTG	AGTGACAATGTGAGGCAATTAG	182 bp
** *IL-6* **	GTACATCCTCGACGGCATC	ACCTCAAACTCCAAAAGACCAG	198 bp
** *TIMP1* **	CCAAGCCTTAGGGGATGCCG	GCTGTTCCAGGGAGCCACAA	175 bp
** *MMP-9* **	GTCTTCCCCTTCACTTTCCTG	GAGGAATGATCTAAGCCCAGC	197 bp
** *MMP-2* **	CAGTGACGGAAAGATGTGGT	TGGTGTAGGTGTAAATGGGTG	182 bp
** *MMP-13* **	GTTGCTGCGCATGAGTTCGG	AACCTGCTGAGGATGCAGGC	294 bp
** *RPS18* **	CTTTGCCATCACTGCCATTAAG	TCCATCCTTTACATCCTTCTGTC	199 bp

## Data Availability

Not applicable.

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
