# Peer review of "Evaluation of Cell Migration and Cytokines Expression Changes under the Radiofrequency Electromagnetic Field on Wound Healing In Vitro Model"

_ijms, 2022, doi:10.3390/ijms23042205_

Round 1

Reviewer 1 Report

Overview of the manuscript
The manuscript is focused on the use of radiofrequency electromagnetic field (RF-EMF) to explore its therapeutic potential on wound healing, evaluating modulatory effects on inflammatory and re-epithelialization phases. Using three different exposure protocol on cell line HaCat, the authors analyse cytokines expression, oxidative stress, MMP modulation and mechanical stretch model, concluding the useful employ of RF-EMF to promote the healing of wounds.

GENERAL COMMENT

The topic of the work is interesting, and the experimental plan is adequate and well performed to support the conclusion. However, the manuscript suffers of a not good presentation. The introduction should better present the topic related to the mechanism investigated in wound healing.

SPECIFIC COMMENTS

Title

The title is generic and non-specific. The use of radiofrequency that is your real topic, should compare in the title.

Abstract

Pag. 1, line 13-18: the first part of abstract is too generalist, you should focus better on the topic and aim. You have applied radiofrequency to a specific aspect of wound healing this should be clearly highlighted.

Introduction

Pag. 2, line 57: hyperthrophic scars have particular mechanisms of formation, this is not your topic in the present work. Delete the sentence.

Pag. 2, line 41-47: You investigate the expression of MMP. This topic should be better explained in Introduction section.

Pag. 2, line 58-66: The rational of the different protocol of radiofrequency application should be introduced

Pag. 2, line 80-81: the sentence makes more sense as closing remark in discussion sentence.

Results

In several bar-graphs about PCR analysis the CI is very high, and this can hide differences. Do you have any explanation for this? Is gene expression similarly so highly variable in the sham control?

Pag. 4, line 132-134: The paragraph remains difficult to reading. Give indication about mitomycin C. Rewrite it.

Pag. 4, line 155-160: the description of results remains unclear. Protocol A and Protocol B appear to induce a different expression of cytokines. Explain your results better, more schematically.

 Fig. 4, b: Micrograph?? They are bar-graphs!! Explain or correct it.

Fig. 5: Has any significance been found in the bar graph c? The expression of IL-8 gene appears to be  higher than the other cytokines. Could you explain this finding?

 Fig. 6 : the graph a and b are quite different, can you give any explanation on this differences? In graph a, the CI is omitted in the sham bar.

Pag. 6, line 195-202: The explanatory paragraph is very long it should be moved in Introduction section, and here you give only a brief indication. The rational for the use of MMP-13 should be indicated.

 Pag. 6, line 202-203: the sentence is not clear. Rephrase it.

 Pag. 6, line 205: condition A? What is this? Do you mean protocol A?

 Discussion

Throughout the section there are several references without parentheses. Correct it.

Reviewer 2 Report

The study is of interest as there is need for novel methods for treatment of impaired wound.
In this manuscript Authors demonstrated an in vitro study the therapeutic effect of RF-EMF on 
wound healing.
In general, the manuscript is well-written, the abstract is factual and comprehensive, introduction is 
clear and well organized, results and methods is clear, discussion explain what the results mean and 
how they are important.
However, publication need some correction or clarification of some aspects:
1. In my opinion the title is too general, not very precise. The study concerned the therapeutic 
effect of RF-EMF on wound healing. I suggest change it.
2. Section 2.Results:
 where is the Fig.1? I could not find it.
 section 2.5 Figure 6b – first of all I could not find the information how the Authors 
determined the level of IL-6, I supposed Authors used the test ELISA....but I could not find 
this information in section Methods. Also I could not find information how Authors 
prepared the cell culture supernatants. Next, I would like to ask the Authors about the levels 
of IL-6. Authors showed that the production level of IL-6 was between 150 to 210 pg per ml 
of what? Of supernatant? The tested samples were identical? The level of total soluble 
proteins were the same in each samples? Could Authors explain how prepared the 
supernatants?
3. The quality of Fig 4 b, c and Fig 7 is not good enough. The size of font in legend is too 
small.
4. In 3. Discussion and Conclusion there is a little problem with references.
After reading the text of the submitted manuscript, I suggest publishing this work after revision.

Reviewer 3 Report

The article 'Impaired Wound Healing: mechanisms and treatment based on inflammatory response modulation' by Costantini et al. discusses the radiofrequency electromagnetic field (RF-EMF) dependent healing of in vitro keratinocytes wound model. Using a number of in vitro assays, the authors report that RF-EMF treatment promotes keratinocytes' migration and regulates expression of genes involved in healing, such as metalloproteinases (MMPs), tissue inhibitor of metalloproteinases, and pro/anti-inflammatory cytokines, improving wound healing. The study is informative and relevant for the readership of 'IJMS.' The study design and experimental approaches are appropriate, however the study is premature for publication at this stage. Hence, the authors need to address some critical concerns in the current version of the manuscript before its publication. The major and minor issues are listed below.

Major:

How did the authors optimize the doses of RF-EMF?

Figure1 is missing from the manuscript.

Can the authors comment on why H202 levels were significantly different in only Protocol C, whereas GSH levels are the weakest of all three protocols?

Further, the statistics in Figure3b seem misrepresented. The effect size seems small for the significance reported. Can the authors provide box plots for the measurements?

Have the authors checked the status of other ROS, regulators, and ROS- detoxifying enzymes such as SODs, catalases, peroxidases, Peroxiredoxin, Nrf2?  

What was the rationale behind assaying the rate of cell migration at T0, T6, and T24? Since in protocol A, authors decided to expose cells to RF-EMF at 30 min, 6 hr, and 30 min, did it make the rate of cell migration any faster at 30 min post-exposure?

What was the control in gene expression levels in Figure5?

TNFα is generally a pro-inflammatory cytokine. How do the authors think it accelerates the transition from the inflammatory to the healing phase?

Early release of IL-6 promotes pro-inflammatory response, whereas late release induces reparative, anti-inflammatory response. (Johnson et al., Biomedicines, 2020). To distinguish the role of IL-6 between these two extremes, authors should report IL-6 levels both during early and late RF-EMF exposure. 

As the results of individual protocols are not consistent across the assays, it would be helpful if the authors establish the timeline of progression from inflammatory to proliferative phase and subsequently to the remodeling phase in their wound healing assays first and discuss the results of their experiments accordingly. 

The possible explanation for why some experiments demonstrate different results among the protocols (A, B, C) is not convincingly discussed.

Minor:

In multiple places throughout the manuscript, verb-tense inconsistency typographical and grammatical errors are very common. The quality of written English can be improved in the revised version of the manuscript too. Some examples (but not limited to) include:

  1. other than decrease productivity and reduction of work time

other than decrease (decreased) productivity and reduction of work time

  1. the current phase must have the ability to turn-off it and promote the next one.

the current phase must have the ability to turn-off (itself) off  and promote the next one.

  1. Migration epidermal keratinocytes, over the wound site, produce multiple factors to promote re-epithelialization, to stimulate angiogenesis and the generation of connective tissue matrix

Migration (of) epidermal keratinocytes, over the wound site, produce multiple factors to promote re-epithelialization, to stimulate angiogenesis and the generation of connective tissue matrix.

  1. Cells were pleated on three different exposure conditions, each with its own sham control (Fig. 8).

Cells were pleated (plated) on three different exposure conditions, each with its own sham control (Fig. 8).

  1. Line 183: in accord with IL-6 ability to switch from inflammatory to a reparative phase.

Line 183: in accord (accordance) with IL-6 ability to switch from inflammatory to a reparative phase. Also provide reference here.

Please provide scale bars in images in Figure 4.

Line 198: High levels of MMP-2 andcMMP-9 and low levels of Tissue Inhibitors of metalloproteinase (TIMP1) are present in chronic wounds, and their inhibition can help to improve healing process. MMP-9 creates a state of persistent inflammation and tissue destruction by digestion of fibronectin in fragments, but if the levels of MMP-9 are lower than normal, epithelialization may be wavered.

Please provide appropriate references.

References are not formatted consistently at some places in the manuscript.

Round 2

Reviewer 3 Report

The revised version of the manuscript 'Evaluation of cell migration and cytokines expression changes under the radiofrequency electromagnetic field on wound healing in vitro model' by Costantini et al. has addressed all the relevant major concerns raised in the previous version of the paper. The manuscript is now mostly fit for publication in 'International Journal of Molecular Sciences'. I endorse the publication of this paper in the journal; however, the authors should address two minor concerns in the manuscript before final publication.

The authors are recommended to include the following sentence in the methods/results section of the manuscript:

'In accord with the timing of the in vivo application of RF-EMF devices, our objective was to evaluate intermittent, acute, or protracted exposure.'

The authors quote that 'Later TNFα also promotes epithelial regeneration by activation of Notch pathway'; however, they have not tested any Notch target genes in their system. Either they should exclude Notch and propose NF-kB-dependent IL-6 activation or examine some Notch target genes. 
